# Correlation between ADC Histogram-Derived Metrics and the Time to Metastases in Resectable Pancreatic Adenocarcinoma

**DOI:** 10.3390/cancers14246050

**Published:** 2022-12-08

**Authors:** Riccardo De Robertis, Luisa Tomaiuolo, Francesca Pasquazzo, Luca Geraci, Giuseppe Malleo, Roberto Salvia, Mirko D’Onofrio

**Affiliations:** 1Department of Radiology, Ospedale G.B. Rossi, University of Verona, 37134 Verona, Italy; 2Department of Pancreatic Surgery, Ospedale G.B. Rossi, University of Verona, 37134 Verona, Italy

**Keywords:** pancreas, radiomics, histogram analysis, magnetic resonance imaging, metastasis, pancreatic carcinoma

## Abstract

**Simple Summary:**

Local and systemic relapse within the first year after curative surgery occurs in up to 60% of patients with pancreatic ductal adenocarcinoma (PDAC). An improvement in the preoperative prognostic stratification would be clinically beneficial to spare patients from unbeneficial upfront surgery, and to optimize the follow-up. The aim of this study was to correlate conventional magnetic resonance features and the metrics derived from the histogram analysis of apparent diffusion coefficient (ADC) maps, with the risk and the time to metastases (TTM) in patients with resectable PDAC. The ADC skewness had a significant effect on the risk of metastases (hazard ratio = 5.22, *p* < 0.001). Patients with an ADC skewness ≥0.23 had a significantly shorter TTM than those with a skewness <0.22 (11.7 vs. 30.8 months, *p* < 0.001). Histogram analysis of ADC maps provides parameters correlated to the metastatic potential of PDAC; higher ADC skewness seems to be associated with a significantly shorter TTM.

**Abstract:**

Background: A non-invasive method to improve the prognostic stratification would be clinically beneficial in patients with resectable pancreatic adenocarcinoma (PDAC). The aim of this study was to correlate conventional magnetic resonance (MR) features and the metrics derived from the histogram analysis of apparent diffusion coefficient (ADC) maps, with the risk and the time to metastases (TTM) after surgery in patients with PDAC. Methods: pre-operative MR examinations of 120 patients were retrospectively analyzed. Patients were grouped according to the presence (M+) or absence (M−) of metastases during follow-up. Conventional MR features and histogram-derived metrics were compared between M+ and M− patients using the Fisher’s or Mann–Whitney tests; receiver operating characteristic (ROC) curves were constructed for the features that showed a significant difference between groups. A Cox regression analysis was performed to identify the features with a significant effect on the TTM, and Kaplan–Meier curves were constructed for significant features. Results: 68.3% patients developed metastases over a mean follow-up time of 29 months (range, 3–54 months). ADC skewness and kurtosis were significantly higher in M+ than in M− patients (*p* < 0.001). Skewness had a significant effect on the risk of metastases (hazard ratio—HR = 5.22, *p* < 0.001). Patients with an ADC skewness ≥0.23 had a significantly shorter TTM than those with a skewness <0.22 (11.7 vs. 30.8 months, *p* < 0.001). Conclusions: pre-operative histogram analysis of ADC maps provides parameters correlated to the metastatic potential of PDAC. Higher ADC skewness seems to be associated with a significantly shorter TTM in patients with resectable PDAC.

## 1. Introduction

Pancreatic ductal adenocarcinoma (PDAC) is diagnosed at an advanced stage of disease in more than 50% of cases and, despite several improvements in the multimodality treatment strategies, the 5-year relative survival is lower than 15% [1]. Less than 20% of patients are diagnosed early enough for upfront curative-intent surgical resection [1]. Improvements in adjuvant treatments had a survival benefit in patients with resected PDAC [2], even though a significant proportion of patients experience local and systemic metastases, and eventually die within the first 12 months after curative-intent surgery [3]. One of the biggest challenges in patients with resectable PDAC is to assess the individual prognosis to avoid unbeneficial surgery in those with a high risk of early disease recurrence. The reasons for early disease recurrence in resectable PDAC are not completely understood and probably involve genomic and histological tumor heterogeneity that translates into aggressive biological behavior, with presence of micrometastases undetectable at imaging even for resectable lesions; unfortunately, genomic features of PDAC can be assessed only on histological samples from biopsy, which is not usually performed on resectable PDACs diagnosed by imaging, and resection specimens [4,5,6]. Previous studies [7,8] identified some imaging features of PDAC that may serve as prognostic biomarkers, differentiating patients at high risk of early disease recurrence from those who are not, but this approach has limited value in clinical practice. Non-invasive methods that could improve the pre-operative prognostic stratification of these patients would be, therefore, clinically beneficial. Radiomics is research field of radiology based on the assumption that biomedical images contain information that reflects inner histopathological features of solid tumors; these are imperceptible by the human eye, but they can be quantified and described through the mathematical extraction of metrics describing the spatial distribution, and the interrelationships between the pixels composing the images. Histogram analysis is a simplified radiomic approach that evaluates the distribution of grey levels, within a region of interest (ROI), drawn on a biomedical image using descriptive parameters called first-order statistics. These can provide great insights into tumor microenvironment and heterogeneity related to cellular morphology and density, metabolism, angiogenesis, and gene expression [9]. Changes in histogram metrics, shape and asymmetry reflect microstructural and functional differences in tumor composition correlated to biological aggressiveness and prognosis; these might be of relevant interest to develop targeted therapeutic strategies for cancer patients, for example to identify those patients that could benefit from preoperative chemotherapy rather than upfront surgery. The predictive and prognostic role of histogram-derived parameters was confirmed by previous studies, which demonstrated their usefulness in the identification of pancreatic neuroendocrine neoplasms (panNENs) with higher biological aggressiveness and worse prognosis, and to assess the malignant potential of pancreatic intraductal papillary mucinous neoplasms (IPMNs) [10,11,12,13,14,15]. Histogram analysis also demonstrated a potential role in predicting recurrence-free survival after surgical resection in patients with PDAC [14]. The aim of this study was to correlate conventional magnetic resonance (MR) features and the metrics derived from the histogram analysis of apparent diffusion coefficient (ADC) maps, with the risk and the time to metastases (TTM) in patients with PDAC receiving upfront curative-intent surgery.

## 2. Materials and Methods

One-hundred and forty-one patients with treatment-naïve, resectable PDAC who underwent MR imaging before surgery between 2013 and 2021 were retrospectively identified. Inclusion criteria were: (a) surgical resection within 1 month from MR imaging; (b) optimal diagnostic quality of MR images; (c) absence of biliary stents or drainage tubes; (d) at least 3 months of clinical and radiological post-operative follow-up. Exclusion criteria were: (a) surgery performed more than 1 month from MR; (b) presence of image artifacts; (c) previous biliary drainage; (d) follow-up shorter than 3 months. Pathological and clinical data were retrospectively retrieved by patients’ medical records. Metastases were diagnosed by means of follow-up imaging (computed tomography—CT and/or MR imaging), together with CA 19.9 dosage, and physical examination.

MR examinations were performed using a 1.5 T unit (Ingenia, Philips Medical Systems, Eindhoven, The Netherlands; or MAGNETOM Avanto, Siemens Healthineers, Erlangen, Germany), using a multi-channel phased-array torso coil. Pre-contrast imaging included T1- and T2-weighted images and diffusion-weighted imaging (DWI); ADC maps were automatically reconstructed with a monoexponential decay model from a DWI echo-planar imaging single-shot (EPI-SS) sequence based on 3 b-values (0, 400, and 800 s/mm^2^). DW images were acquired with a slice thickness of 5 mm during free breathing. Post-contrast images were acquired after the administration of 0.1 mmol/kg of body weight of gadopentate dimeglumine (Multihance; Bracco, Milan, Italy) at an injection rate of 1.5–2 mL/s. The timing for post-contrast imaging was determined by fixed delays (30–45 s after the start of contrast medium administration for arterial phase images; 60–70 s for portal phase images; and >180 s for delayed phase images). MR cholangiopancreatography (MRCP) images were acquired in all patients.

Each MR examination was reviewed by a Radiologist with 10 years of experience in abdominal imaging, who evaluated the image quality, and the following conventional features: tumor location (head, body, or tail); tumor size; signal intensity on T1- and T2-weighted images (hypointense, isointense, or hyperintense) compared to the adjacent pancreatic parenchyma; tumor enhancement on arterial, portal, and delayed phases. Tumor segmentation and data extraction for histogram analysis were performed using a software for medical image processing (Lifex; www.lifexsoft.org, accessed on 1 March 2022) [16]. Tumors were segmented semi-automatically on the ADC map by taking into account all of the pulse sequences to delineate the tumor; three-dimensional volumes of interest (VOIs) were automatically obtained. For feature extraction the following parameters were set, as previously reported [17]: spatial resampling 2 × 2 × 2 mm; intensity discretization at 64 gray levels; intensity rescaling at 64 gray levels, between the absolute minimum and maximum values in the VOI. The following histogram metrics were extracted: minimum value (ADCmin), maximum value (ADCmax), mean value (ADCmean), median value (ADCmedian), standard deviation (SD), skewness, kurtosis, entropy, uniformity, 25th percentile (ADC25), 75th percentile (ADC75). Skewness is the measure of the asymmetry of a histogram and is calculated as:$$\frac{\frac{1}{N_{v}}\sum_{k=1}^{N_{v}}{\left(X_{d,k}-\mu \right)}^{3}}{{\left(\frac{1}{N_{v}}\sum_{k=1}^{N_{v}}{\left(X_{d,k}-\mu \right)}^{2}\right)}^{3/2}}$$
where *μ* is the mean of the distribution histogram, *N_v_* is the number of voxels in the ROI, and *X_d_* is the set of *N_g_* discretized intensities of the *N_v_* voxels in the ROI [18]. Kurtosis is a measure of the “tailedness” of the distribution of a variable and represents a combined weight of a distribution’s tails to the center of the distribution. According to [18], kurtosis of a histogram is calculated as:$$\frac{\frac{1}{N_{v}}\sum_{k=1}^{N_{v}}{\left(X_{d,k}-\mu \right)}^{4}}{{\left(\frac{1}{N_{v}}\sum_{k=1}^{N_{v}}{\left(X_{d,k}-\mu \right)}^{2}\right)}^{2}}-3$$
where *μ* is the mean of the distribution histogram, *N_v_* is the number of voxels in the ROI, and *X_d_* is the set of *N_g_* discretized intensities of the *N_v_* voxels in the ROI. The harmonization of the radiomic features between the two scanners was performed by applying the ComBat algorithm [19].

Patients were grouped according to the presence or the absence of metastases during the follow-up period (M+ vs M−). Conventional MR features and histogram-derived parameters were compared between groups using Fisher’s or Mann–Whitney U tests. Receiver operating characteristic (ROC) curves were constructed for the features that showed a significant difference between groups; to evaluate their performance in identifying M+ patients, the areas under the ROC curves (AUCs) were calculated. Optimal cut-off points were then identified through the calculation of the Youden’s index [20] and sensibility, specificity, positive and negative predictive values, and accuracy of the features were calculated. The ROC curves were then compared with the method described by Delong et al. [21]. Time to metastases was defined as the time that elapsed from the day of surgery until the onset of metastases; patients without an event were censored at the time of their last follow-up. The Cox regression model was used to identify the variables with a significant effect on the development of metastases; the numerical variables, such as the histogram metrics, were dichotomized (i.e., high risk vs. low risk) to be analyzed with the Cox regression model by constructing their ROC curves, and calculating optimal cut-offs according to the Youden’s index. Kaplan–Meier curves were finally constructed for the parameters that showed a significant effect on the risk of metastases; differences between the Kaplan–Meier curves were assessed using the Log-rank test. *p* values ≤ 0.05 were statistically significant. Statistical analysis was performed using SPSS, v. 23 (IBM, Chicago, IL, USA).

## 3. Results

### 3.1. Study Population

Twenty-one patients were excluded from this study owing to a suboptimal image quality (n = 1), presence of a biliary stent/drainage tube (n = 10), or a post-operative follow-up shorter than 3 months (n = 10). Finally, 120 patients with a mean age of 65 years were included in this study. The mean follow-up length was 29 months (range, 3–54 months). Metastases occurred in 82/120 patients (68.3%). Baseline and follow-up demographic and clinical features of the study population are presented in detail in Table 1.

### 3.2. Image Analysis

The results of the Fisher’s and the Mann–Whitney U tests for comparison of qualitative and quantitative tumor features are presented in Table 2 and Table 3.

The mean tumor size was 28 mm (range 7–60 mm). Most tumors were located in the pancreatic head (82.5%) and were T1-hypointense (97.5%), and T2-hyperintense (64.2%). Hypoenhancement on the three post-contrast phases was most frequent (94.2%, 91.7%, and 86.6%). None of the conventional MR features showed a significant difference between M+ and M− patients (all *p* > 0.05). Among the histogram-derived parameters, skewness and kurtosis were significantly higher in M+ than M− patients (0.6 vs. 0.2 and 4.3 vs. 3.8, *p* = 0.005 and 0.032, respectively).

Two examples are presented in Figure 1 and Figure 2.

The ROC curves of ADC skewness and kurtosis for identification of M+ patients are presented in Figure 3. ADC skewness had higher AUC than ADC kurtosis for the identification of M+ patients, although no significant differences were found between the ROC curves (0.754 vs. 0.724; *p* = 0.89).

The diagnostic values of ADC skewness and kurtosis in identifying M+ patients in terms of sensitivity, specificity, positive and negative predictive values, and accuracy are presented in Table 4.

### 3.3. Correlation with the TTM

Over a mean follow-up time of 29 months (range 3–54 months) 82 patients developed distant metastases (68.3%): 57 (69.5%) only in the liver, 16 (19.5%) in multiple sites, and 9 (11%) only in the lungs. The mean TTM was 11.4 months (range 2–38 months). At Cox regression analysis, the only variable that showed a significant effect on the risk of development of metastases was ADC skewness (hazard ratio—HR = 5.22, *p* < 0.001). Patients with an ADC skewness ≥0.23 had a significantly shorter TTM than those with a skewness <0.22 (11.7 vs. 30.8 months, *p* < 0.001); the Kaplan–Meier curve is presented in Figure 4.

## 4. Discussion

To the best of our knowledge this is the first study that has evaluated the potential role of ADC histogram analysis of resectable PDAC to predict the development of distant metastases after surgery. Surgery remains the only potential curative treatment for PDAC patients, even though it is burdened by a significant rate of early postoperative disease progression, and high complication rates. There is, therefore, the clinical need to better stratify patients’ prognosis preoperatively to select only those that could really benefit from upfront surgical resection, and to spare patients from complications of an unbeneficial surgery. Our results suggest that histogram analysis of ADC maps may provide non-invasive biomarkers that could be helpful to better stratify PDAC patients’ prognosis before surgery. Histogram analysis is a radiomic technique that evaluates the distribution of grey levels within a ROI representative of the tumor mass on CT or MR images, deriving several first-order metrics that describe the frequency of pixels exhibiting the same grey level, and thus providing information on tumor microarchitecture and heterogeneity. By applying this method to ADC maps, it is possible to create a histogram that describes the distribution of the ADC values within the ROI. Previous studies reported that histogram metrics derived from ADC maps can explore inner pathological features of pancreatic tumors, with promising results in terms of better characterization and non-invasive prognostication. Lu et al. [13] found that the 75th percentile value of ADC1000 had an AUC of 0.781 and a sensitivity of 91% for differentiating intestinal- from pancreatobiliary-type periampullary adenocarcinoma, even though specificity was 59%. Shindo et al. [23] reported that ADC histogram-derived metrics were useful to distinguish between PDAC and pancreatic neuroendocrine neoplasms (pNENs): the mean ADC200 and ADC400 were significantly higher in pNENs than in PDACs (*p* = 0.001 and 0.019, respectively); PDACs showed significantly higher skewness and kurtosis on ADC400 (*p* = 0.007 and 0.001, respectively) and ADC800 (*p* = 0.001); with all b-value combinations, the ADC entropy was significantly higher in PDACs, and showed the highest AUC for diagnosing this histotype. The most intriguing role of ADC histogram analysis is the prediction of the biological behavior of pancreatic tumors. In this regard, previous studies reported that several ADC histogram metrics may be able to identify tumors with adverse pathological features and worse prognosis [10,11,12,13,14,15]. Pereira et al. [10] reported a significant correlation between the tumor grade of pNENs, and several histogram-derived parameters. In particular, the mean ADC, 75th, 90th, and 95th percentiles were significantly higher in G1 tumors compared to G2 and G3 tumors; skewness and kurtosis were significantly different between G1 and G3 tumors; while no statistically significant differences were found between G2 and G3 tumors. Another study [12] reported that ADC entropy was significantly higher in G2-3 pNENs, with AUC of 0.757; sensitivity and specificity were 83.3% and 61.1%; ADC kurtosis was higher in pNENs with vascular involvement, nodal and hepatic metastases (*p* = 0.008, 0.021 and 0.008; AUC = 0.820, 0.709 and 0.820); sensitivity and specificity were 85.7/74.3%, 36.8/96.5%, and 100/62.8%. Hoffman et al. [11] reported that the whole-lesion ADC entropy (5.1 ± 0.2 vs. 5.4 ± 0.2; *p* = 0.01, AUC = 0.86), mean of the bottom 10th percentile (2.2 ± 0.4 vs. 1.6 ± 0.7; *p* = 0.03; AUC = 0.81), and mean of the 10–25th percentile (2.8 ± 0.4 vs. 2.3 ± 0.6; *p* = 0.04; AUC = 0.79) demonstrated significant differences between benign and malignant IPMNs. ADC entropy was the highest performing histogram metric and achieved a sensitivity of 100%, a specificity of 70%, and an accuracy of 83% for predicting malignancy. At multivariable analysis of ADC histogram metric and conventional MR features, entropy was the only significant independent predictor of malignancy in IPMNs (*p* = 0.004). Igarashi et al. [15] found that ADC entropy could predict high-grade dysplasia in IPMNs with a diagnostic accuracy of 73%. Noda et al. [24] reported that ADC kurtosis, entropy, and energy were significantly associated with overall survival in PDAC patients; the ADC kurtosis had the highest AUC for predicting 3-year survival (0.824) among these three parameters, and lower survival rates occurred in patients with kurtosis >2.45 compared to those with lower kurtosis values (*p* < 0.001). Another study [14] reported that tumor differentiation, the nodal ratio, and the ADCmax value were significant predictors of recurrence-free survival after resection in patients with PDAC; in the same study, tumor differentiation, ADC uniformity and arterial entropy were significant predictors of death, with HR of 2.82, 3.32, and 6.84, and patients with higher arterial entropy had significantly shorter overall survival than other patients (*p* = 0.01, median 24 vs. 31 months).

In the present study, ADC skewness was a significant predictor of the development of metastases in patients with resectable PDAC. According to the Cox regression analysis, patients with a skewness greater than 0.23 had a risk of metastases more than five times greater than patients with a skewness value below this cut-off. Moreover, a higher skewness was correlated with a significantly shorter TTM (11.7 vs. 30.8 months, *p* < 0.001). ADC skewness may, therefore, be a radiological biomarker of higher biological aggressiveness in PDAC. Skewness reflects the asymmetry of the ADC distribution histogram; if a histogram has an elongated tail on the left side of the mean, it is negatively skewed, while if the tail is elongated on the right side of the mean, it is positively skewed. Previous studies demonstrated a significant correlation between this parameter and several structural, physiological, molecular, and metabolic characteristics of solid tumors; therefore, this makes skewness a reliable marker of heterogeneity in several solid tumors, as gliomas and glioblastomas, endometrial and cervical cancer, renal cell carcinoma, and head and neck squamous cell carcinoma [9].

Our study had several limitations. First, the validity of our results may be limited by a relatively short follow-up period, with a mean of 29 months; this is a consequence of the natural history of PDAC patients, whose 5-year survival rate after surgery barely reaches 20%. Second, data on pre-operative CA 19.9 levels and performance status, which are known to be important prognostic factors in resectable pancreatic cancer [25], as well as those on adjuvant chemotherapy, were not available for most of the patients within the study population. In most cases they were referred to our specialized center for surgery, and were discharged and referred to peripheral centers for follow-up and adjuvant treatments. Nevertheless, even though several trials reported that adjuvant chemotherapy may prolong the disease-free survival and metastasis-free survival in resected PDAC patients [2,26], things are different in the “real world”; up to 30% of patients do not receive all the planned cycles of adjuvant chemotherapy, within a short interval from surgery, because of comorbidities, the worsening of performance status, and post-operative complications [27]. Therefore, the real benefit of adjuvant therapy is not clearly established. Third, tumor segmentation was performed by one radiologist and our results were not tested on a control population, therefore the reproducibility of the measurements and the applicability of our results were not assessed; an external and prospective validation of our results would be useful.

## 5. Conclusions

Histogram analysis of ADC maps provides metrics that correlate to the metastatic potential of resectable PDAC. Our results suggest that ADC skewness could be a valid biomarker in resectable PDAC patients, as it could be used to predict the development of metastases after surgery, thereby improving prognostic stratification. If our results are confirmed by further studies, histogram-derived parameters could be integrated into the decisional algorithms, and may play an important role in identifying PDAC patients that could really benefit from upfront surgery.

## Figures and Tables

**Figure 1 cancers-14-06050-f001:**
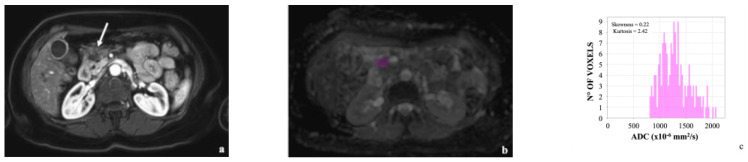
(**a**) 59-year-old woman with pancreatic head carcinoma (arrow); (**b**) tumor ROI is shown in purple; (**c**) histogram analysis revealed a skewness of 0.22 and kurtosis of 2.42, respectively. No metastases occurred over a 15-month follow-up after pancreaticoduodenectomy.

**Figure 2 cancers-14-06050-f002:**
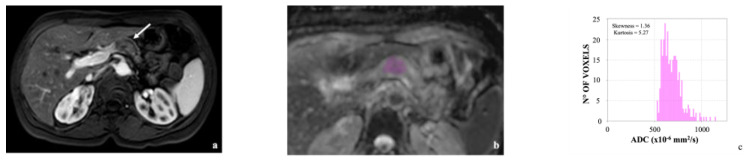
57-year-old man with pancreatic body carcinoma (arrow in (**a**), purple ROI in (**b**)); (**c**) histogram analysis resulted in a skewness of 1.36 and a kurtosis of 5.27. Hepatic metastases developed 12 months after distal pancreatectomy.

**Figure 3 cancers-14-06050-f003:**
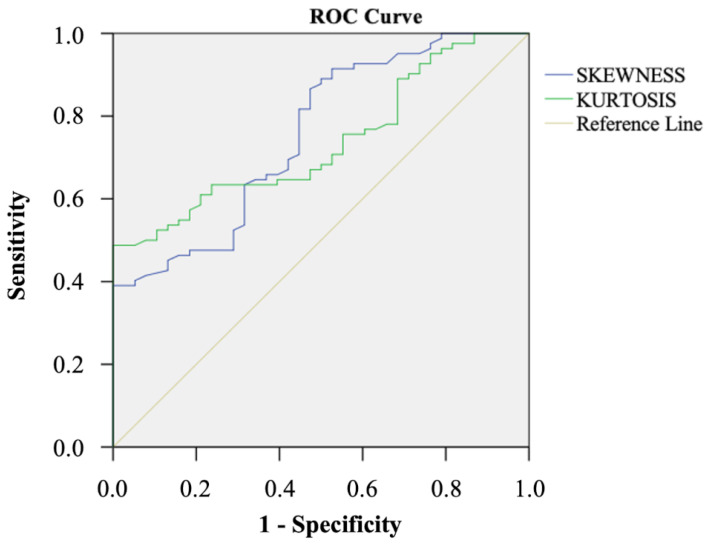
Comparison of the ROC curves of skewness and kurtosis for identification of M+ patients.

**Figure 4 cancers-14-06050-f004:**
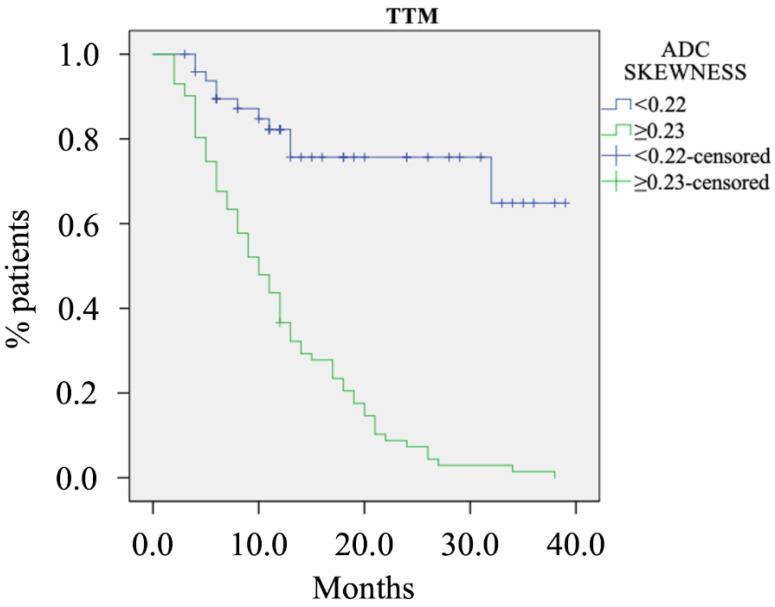
Kaplan–Meier curve for ADC skewness. Legend: TTM, time to metastases.

**Table 1 cancers-14-06050-t001:** Demographic and clinical features of the study population; data are number of cases (%), except where specified. T and N parameters and tumor stage are expressed according to the American Joint Committee on Cancer (AJCC) Cancer Staging Manual, 8th edition [22].

Features	
Sex	
Male	72 (60%)
Female	48 (40%)
Age ^a^ (years)	65 (42–86)
pT1b	1 (0.8%)
pT1c	40 (33.3%)
pT2	62 (51.7%)
pT3	17 (14.2%)
pN1	5 (4.2%)
pN2	115 (95.8%)
Tumor stage	
IIB	5 (4.2%)
III	115 (95.8%)
Follow-up ^a^ (months)	29 (3–54)
Metastases	
Yes	82 (68.3%)
Liver	42 (51.2%)
Lung	10 (12.2%)
Other sites	8 (9.8%)
Multiple sites	22 (26.8%)
No	38 (31.7%)

^a^ mean (range). Legend: TTM, time to metastases. pT, pathological T stage; pN, pathological N stage.

**Table 2 cancers-14-06050-t002:** Results of the Fisher’s exact test for comparison of conventional MR features; data are number of cases (%).

Feature	Total	M+	M−	*p*
**Site**				0.308
Head	99 (82.5%)	70 (85.4%)	29 (76.3%)	
Body	19 (15.8%)	11 (13.4%)	8 (21.1%)	
Tail	2 (1.7%)	1 (1.2%)	1 (2.6%)	
**T1w SI**				1
Hypointense	117 (97.5%)	80 (97.6%)	37 (97.4%)	
Isointense	3 (2.5%)	2 (2.4%)	1 (2.6%)	
**T2w SI**				0.399
Hypointense	11 (9.2%)	7 (8.5%)	4 (10.5%)	
Isointense	32 (26.6%)	25 (30.5%)	7 (18.4%)	
Hyperintense	77 (64.2%)	50 (61%)	27 (71.1%)	
**Arterial phase SI**				0.678
Hypointense	113 (94.2%)	78 (95.1%)	35 (92.1%)	
Isointense	7 (5.8%)	4 (4.9%)	3 (7.9%)	
**Portal phase SI**				0.242
Hypointense	110 (91.7%)	77 (93.9%)	33 (86.8%)	
Isointense	9 (7.5%)	4 (4.9%)	5 (13.2%)	
Hyperintense	1 (0.8%)	1 (1.2%)	0 (0%)	
**Delayed phase SI**				0.487
Hypointense	104 (86.6%)	73 (89%)	31 (81.6%)	
Isointense	11 (9.2%)	6 (7.3%)	4 (10.5%)	
Hyperintense	5 (4.2%)	3 (3.7%)	3 (7.9%)	

Legend: T1w, T1-weighted; T2w, T2-weighted; SI, signal intensity.

**Table 3 cancers-14-06050-t003:** Results of the Mann–Whitney U test for comparison of quantitative conventional MR features and histogram-derived parameters; data are mean (range).

Parameter	M+	M−	*p*
Age	65 (42–86)	66 (46–83)	0.66
Size	27.5 (10–58)	28.4 (7–60)	0.81
ADC_min_	677.3 (1–1541)	666.2 (16–1206)	0.95
ADC_max_	2363 (1049–3607)	2164 (249–3541)	0.10
ADC_mean_	1361.6 (658–1881)	1341.9 (175–1875)	0.99
SD	295.1 (35–848)	280.4 (24–707)	0.52
ADC_median_	1329.4 (652–1871)	1320.6 (177–1831)	0.80
ADC_25_	1157.4 (18;1793)	1157.7 (165–1587)	0.74
ADC_75_	1529.9 (725;2124)	1509.6 (190–2202)	0.82
Skewness	0.6 (−0.6;3.3)	0.2 (−1.2;1.8)	0.005
Kurtosis	4.3 (1.7; 17.3)	3.8 (2.1; 11.1)	0.032
Entropy	6.5 (1.3–9.3)	6.4 (1.2–9.4)	0.31
Uniformity	0.1 (0–0.1)	0.1 (0–0.4)	0.36

Legend: M+, metastatic patients; M−, non-metastatic patients; ADC_min_, minimum ADC value; ADC_max_, maximum ADC value; ADC_mean_, mean ADC value; ADC_median_, median ADC value; SD, standard deviation; ADC_25_, 25th percentile; ADC_75_, 75th percentile.

**Table 4 cancers-14-06050-t004:** Diagnostic values of skewness and kurtosis in identifying M+ patients.

	ADC Skewness	ADC Kurtosis
Optimal Cut-off	0.23	3.90
Sensitivity	98.6 (92.5–100)	47.6 (36.4–58.9)
Specificity	41.7 (27.6–56.8)	100 (91–100)
PPV	71.7 (66.6–76.3)	100 (-)
NPV	95.2 (73.5–99.3)	46.9 (41.8–52.1)
Accuracy	75.8 (67.2–83.2)	64.2 (54.9–72.7)

Legend: PPV, positive predictive value; NPV, negative predictive value.

## Data Availability

Not applicable.

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
