# Peer review of "Correlation between ADC Histogram-Derived Metrics and the Time to Metastases in Resectable Pancreatic Adenocarcinoma"

_cancers, 2022, doi:10.3390/cancers14246050_

Round 1

Reviewer 1 Report

Summary

De Robertis et al. analyzed the significance of apparent diffusion coefficient (ADC) value for prognosis in resectable pancreatic adenocaricnoma. Although the authors showed the significance of ADC histogram analysis for predicting time to metastasis in pancreatic carcinoma, there are some major points to be revised.

Major points:

1)     Results [Study population] Page 3 line 134; In the table 1, the authors seemed to show that the patients background demographic data at the time of data cut-off, because the metastatic sited has been presented in the table1. However, the background data should be presented at the time of diagnosis. The authors should also show the T, N, M factors and stage (according to the UICC 8th edition).

2)     Results [Image analysis] Page 5 line 144; In the table2, the authors compared the conventional MRI features between M+ and M- patients. The authors should show the overall survival and recurrence free survival data of the M+ and M- patients. The authors should also explain the contents of the table 2 in the main text to get readers more easily to understand.

3)     Results [Image analysis] Page 6 line 150; This table named “Table 2” should be renamed as “Table 3.”

4)      Results [Correlation with the TTM] Page 10 line 198; If the authors would want to show ADM skewness is significantly correlated with TTM, the authors should more specific patients’ characteristics data. Specifically, preoperative CA19-9, performance status and adjuvant/neoadjuvant chemotherapy and their regimen should be presented. These are known important prognostic factors in resectable pancreatic cancer (Yamaguchi et al. Ann Surg. 2022 Jun 1;275(6):1043-1049; Wu et al. BMC Cancer. 2021 May 27;21(1):624.)

Minor points:

1)     Introduction Page2 line46; The authors stated that the pancreatic cancer could be deadly within 1 year from curative intent surgery. However, in the recent studies, they showed that the 1 year overall survival after curative surgery became more favorable prognosis (SEER Cancer Stat Facts: Pancreatic Cancer: https://seer.cancer.gov/statfacts/html/pancreas.html; Yamaguchi et al. Ann Surg. 2022 Jun 1;275(6):1043-1049.) Please reconsider the text.

Author Response

Thank you very much for Your review.

Below You can find a detailed point-to-point response to Your suggestions; Language editing has been performed.

1) Results [Study population] Page 3 line 134; In the table 1, the authors seemed to show that the patients background demographic data at the time of data cut-off, because the metastatic sited has been presented in the table1. However, the background data should be presented at the time of diagnosis. The authors should also show the T, N, M factors and stage (according to the UICC 8th edition).

Please note that table 1 has been revised according to Your suggestions: pT and pN factors as well as tumor stage accoridng to the AJCC 8th ed are now reported. No patient had metastases at the time of surgery.

2)     Results [Image analysis] Page 5 line 144; In the table2, the authors compared the conventional MRI features between M+ and M- patients. The authors should show the overall survival and recurrence free survival data of the M+ and M- patients. The authors should also explain the contents of the table 2 in the main text to get readers more easily to understand.

Table 2 reports the results of the Fisher's test for comparison of conventional MR features between patients who developed metastases and those who did not. A brief explanation of the findings reported in this table is reported just above the table. In this study we did not aim to evaluate overall survival and recurrence free survival, therefore these data are not provided.

3)     Results [Image analysis] Page 6 line 150; This table named “Table 2” should be renamed as “Table 3.”

Ok, corrected.

4)      Results [Correlation with the TTM] Page 10 line 198; If the authors would want to show ADM skewness is significantly correlated with TTM, the authors should more specific patients’ characteristics data. Specifically, preoperative CA19-9, performance status and adjuvant/neoadjuvant chemotherapy and their regimen should be presented. These are known important prognostic factors in resectable pancreatic cancer (Yamaguchi et al. Ann Surg. 2022 Jun 1;275(6):1043-1049; Wu et al. BMC Cancer. 2021 May 27;21(1):624.)

This is a radiological study that aimed to investigate MRI features in predicting the risk of metastases, and, as such, several clinical and biochemical data were not incldued in out analysis; this is a potential limitation of our study as stated in the Discussion. The citation of Wu et al. BMC Cancer. 2021 May 27;21(1):624 has been added in the Discussion. The article by Yamaguchi et al. Ann Surg. 2022 Jun 1;275(6):1043-1049 is about neoadjuvant chemotherapy in borderline-resectable PDAC, while our study is focused on resectable PDAC that usually do not receive preoperative chemotherapy.

Minor points:

1)     Introduction Page2 line46; The authors stated that the pancreatic cancer could be deadly within 1 year from curative intent surgery. However, in the recent studies, they showed that the 1 year overall survival after curative surgery became more favorable prognosis (SEER Cancer Stat Facts: Pancreatic Cancer: https://seer.cancer.gov/statfacts/html/pancreas.html; Yamaguchi et al. Ann Surg. 2022 Jun 1;275(6):1043-1049.) Please reconsider the text.

Introduction has been modified accordingly

Reviewer 2 Report

The authors have described a method using histogram analysis to predict the rate of metastases after curative-intent surgery. Experiments are well designed, and results are well analyzed. The manuscript has discussed the authors' excellent work and is appropriate to be published on Cancers in the present form. 

Author Response

Thank you very much for Your review.

Please note that language editing has been performed.

Reviewer 3 Report

The authors present a manuscript using analysis of magnetic resonance images in an effort to better define which patients would likely develop metastasis and therefore might not benefit from surgical resection. Pancreatic cancer has a dismal survival rate and studies investigating potential to identify metastatic patients will provide critical resources for the field. While this paper demonstrates a correlation of skewness and kurtosis from ADC analysis, the rigor of the analysis is relatively low. There are multiple issues that should be addressed for this to be acceptable.

Major issues:

It is critical to demonstrate histologically in your study that your cutoff point for the ACD correlates with actual differences in the histological features and/or grade through HandE or other staining analysis. This could be done on banked paraffin samples or alternative proof of principle could be done with any PDAC samples that also have MR. This would be even better if correlated with transcriptomic data and PDAC subtype analysis. Without addressing this, the correlation/conclusion to microarchitecture is not supported by the data and a simple biopsy for stage might provide the same result. Tumor staging was also not considered.

Fig1c and Fig2c do not have labeled axis. Additionally for these figures, authors should include plots with all images combined to show the variations between samples demonstrating skewness or kurtosis for M+ as well as M-. An overlay of the average for M+ and M- would also help to demonstrate the differences and potentially assess the relative cutoff values.

Figure 3 should show the same data for M- Patients.

Minor issues:

The skewness and kurtosis should be better defined early in the paper. For many readers it will be unclear what the skewness or Kurtosis is and how it was measured/calculated.

Consideration for tumor location and skewness/kurtosis was not considered. It would be important to determine how/whether location impacts ADC, especially considering the feasibility of resection for non-head tumors.

Author Response

Thank you very much for your detailed review. All you comments were taken into consideration for revision. See below fora detailed point-to-point response.

It is critical to demonstrate histologically in your study that your cutoff point for the ACD correlates with actual differences in the histological features and/or grade through HandE or other staining analysis. This could be done on banked paraffin samples or alternative proof of principle could be done with any PDAC samples that also have MR. This would be even better if correlated with transcriptomic data and PDAC subtype analysis. Without addressing this, the correlation/conclusion to microarchitecture is not supported by the data and a simple biopsy for stage might provide the same result. Tumor staging was also not considered.

The aim of this study was to correlate histogram-derived metrics to a clinical endopoint, namely the time to metastases, rather than to histological findings. Skewness is one of the "agnostic" features  that attempt to capture lesion heterogeneity through quantitative descriptors (see Gillies Radiology 2015).

Fig1c and Fig2c do not have labeled axis.

Labeled axis have been added.

Additionally for these figures, authors should include plots with all images combined to show the variations between samples demonstrating skewness or kurtosis for M+ as well as M-.

This is not clear very clear to me; the images  show two clear examples of M+ and M- tumors with relevant differences in histogram-derived skewness and kurtosis.  

An overlay of the average for M+ and M- would also help to demonstrate the differences and potentially assess the relative cutoff values.

Images have been modified accordingly.

Figure 3 should show the same data for M- Patients.

ROC curves have been designed to identify M+ patients, not M-.

Minor issues:

The skewness and kurtosis should be better defined early in the paper. For many readers it will be unclear what the skewness or Kurtosis is and how it was measured/calculated.

Changes in the text have been performed accordingly; see Materials and methods. 

Consideration for tumor location and skewness/kurtosis was not considered. It would be important to determine how/whether location impacts ADC, especially considering the feasibility of resection for non-head tumors.

Resectability falls out of the aims of this paper, as all tumors were resected. There were no differences between M+ and M- regarding tumor location.

Round 2

Reviewer 1 Report

This paper is a significant contribution, and I think the current revision can be accepted for publication.

Author Response

Thank you very much for your revision.

Kind regards

Reviewer 3 Report

The authors have provided additional data that clarifies the previous concerns. There are some minor revisions suggested.

-Clarification needed on newly added section. Results in lines 231-233 state “Among the histogram-derived parameters, skew-ness and kurtosis were significantly higher in M+ than M- patients (0.77 vs 0.15 and 3.94 vs 2.95, respectively; both p<.001).”  This conflicts with data presented in Table 3 showing M+ of 0.6 vs M- of 0.2 for skewness (p.005) and 4.3 vs 3.8 for Kurtosis (p.032). Please clarify these differences.

-Authors now provide the staging percentages but do not provide data about skewness and kurtosis for those stages. Does the skewness or kurtosis correlate with staging?

-While only two tumor examples are provided (fig 1,2), it would be beneficial to provide the remainder of the ADC histograms as supplementary data. 

-Figure 1 and 2 should be set to the same scale (0-2500).

-Please increase figure text size in Figure 3 and 4.

Author Response

-Clarification needed on newly added section. Results in lines 231-233 state “Among the histogram-derived parameters, skew-ness and kurtosis were significantly higher in M+ than M- patients (0.77 vs 0.15 and 3.94 vs 2.95, respectively; both p<.001).”  This conflicts with data presented in Table 3 showing M+ of 0.6 vs M- of 0.2 for skewness (p.005) and 4.3 vs 3.8 for Kurtosis (p.032). Please clarify these differences.

There was a mistake in the results; sentences in the Results section were changed accordingly.

-Authors now provide the staging percentages but do not provide data about skewness and kurtosis for those stages. Does the skewness or kurtosis correlate with staging?

We did not evaluate the correlation of skewness and kurtosis with tumor stage, as this was not the aim of the study; moreover, with only 5 stage IIB patients the statistical power of such analysis would be very low.

-While only two tumor examples are provided (fig 1,2), it would be beneficial to provide the remainder of the ADC histograms as supplementary data.

Only relevant histograms were saved during image analysis, so most of them are not available for supplementary data; moreover, 120 histograms would be very difficult to read and analyze.

-Figure 1 and 2 should be set to the same scale (0-2500).

Ok changed

-Please increase figure text size in Figure 3 and 4.

Ok changed